# Detection of hepatitis viruses in suspected cases of Viral Haemorrhagic Fevers in Nigeria

**Olumuyiwa Babalola Salu** [1,2]*, **Tomilola Feyikemi Akinbamiro**[2‡], **Remilekun Mercy Orenolu**[1‡], **Onyinye Dorothy Ishaya**[2‡], **Roosevelt Amaobichukwu Anyanwu**[1‡], **Olubunmi Rita Vitowanu**[2‡], **Maryam Abiodun Abdullah**[1‡], **Adenike Hellen Olowoyeye**[2‡], **Sodiq Olawale Tijani**[2‡], **Kolawole Solomon Oyedeji**[1,3‡], **Sunday Aremu Omilabu**[1,2]

1 Centre for Human and Zoonotic Virology, Central Research Laboratory, College of Medicine of the University of Lagos, Idi-araba, Lagos, Nigeria, 2 Department of Medical Microbiology and Parasitology, College of Medicine of the University of Lagos, Idi-Araba, Lagos, Nigeria, 3 Department of Medical Laboratory Science, College of Medicine of the University of Lagos, Idi-Araba, Lagos, Nigeria

☯ These authors contributed equally to this work.
‡ TFA, RMO, ODI, RAA, ORV, MAA, AHO, SOT and KSO also contributed equally to this work.
* osalu@unilag.edu.ng

**Data Availability Statement:** All relevant data are within the manuscript and its Supporting Information files.

## Abstract

There have been several Viral Hemorrhagic Fever (VHF) outbreaks in Nigeria which remains a public health concern. Despite the increasing number of suspected cases of VHF due to heightened surveillance activities and growing awareness, only a few cases are laboratory-confirmed to be VHF. Routinely, these samples are only tested for Lassa virus and Yellow fever virus with occasional testing for Dengue virus when indicated. The aetiology of the disease in these VHF suspected cases in Nigeria which are negative for Lassa, Yellow fever and Dengue viruses remains a puzzle. Since the clinical features exhibited by suspected VHF cases are like other endemic illnesses such as Hepatitis, there is a need to investigate the diversity and co-infections of hepatitis viruses as differentials and possible co-morbidity in suspected cases of VHFs in Nigeria. A total of three hundred and fifty (350) blood samples of 212 (60.6%) males and 138 (39.4%) females, aged <1–70 years with a mean age of 25 ±14.5, suspected of VHFs and tested negative for Lassa, Yellow fever and Dengue viruses were investigated for Hepatitis A, B, C and E viruses at the Centre for Human and Zoonotic Virology (CHAZVY), College of Medicine, University of Lagos (CMUL) using serologic and molecular techniques. The serologic analysis of these VHF suspected cases samples revealed that 126 (36%) were positive for at least one hepatitis virus. Individual prevalence for each of the hepatitis virus screened for showed that 37 (10.6%), 18 (5.1%) and 71 (20.3%) were positive for HBV, HCV and HEV respectively. All the samples were negative for HAV. A co-infection rate of 11.9% was also observed, with HCV/HEV co-infections being the most prevalent and the Northern region having the greatest burden of infection. The evidence of hepatitis virus infections in suspected cases of VHF was documented. Thus, their associations as co-morbidities and/or mortalities in this category of individuals require further investigations in endemic countries such as Nigeria. Therefore, the possible inclusion of screening for hepatitis viruses and other aetiologic agents that could mimic infections in suspected cases of VHFs in Nigeria should be thoroughly evaluated to guide informed policy on the diagnosis and management of these cases.

**Funding:** The author(s) received no specific funding for this work.

**Competing interests:** The authors have declared that no competing interests exist.

## Introduction

Viral Haemorrhagic Fever (VHF) viruses are the major aetiologies of a group of diverse clinical syndromes. The manifestations of these infections range from asymptomatic to life-threatening febrile illnesses with vascular damage and the possibility of haemorrhages in advanced cases [1–3]. However, the incidence of haemorrhage varies extensively between the different viruses [1, 3]. Majorly, VHFs are caused by four taxonomic viral families of enveloped RNA viruses named: *Arenaviridae; Filoviridae; Flaviviridae; Bunyaviridae* [3–6]. The West African sub-region is endemic to several VHFs such as Lassa fever, Ebola haemorrhagic fever, Crimean-Congo haemorrhagic fever, Rift Valley fever, Dengue haemorrhagic fever and Yellow fever [6–9]. Lassa virus the aetiology of Lassa fever is endemic in the countries of Guinea, Sierra Leone, Liberia, Mali, and Nigeria with most of these VHF entities having a zoonotic cycle with rodents and arthropods as the main reservoirs. These VHF agents are transmitted to humans through contact with infected fluids from reservoirs such as faecal matter, saliva, urine, and other body fluids. A few may also be transmitted through arthropod bites [3, 8, 9]. Person-to-person transmission with an incubation period of about 2–21 days has also been documented [10]. The manifestations of the signs and symptoms of the diseases caused by these VHF agents share similarities with most other endemic febrile illnesses in the West African sub-region such as malaria, hepatitis, influenza etc., and differentiating between the agents of these illnesses remains a major challenge [10, 11]. Various diagnostic techniques are available for VHF agents, however, the intensive capital investments as well as highly trained and skilled personnel required remains a major challenge and has limited the availability of diagnostics in resource-limited settings where these agents are endemic [4, 10]. Viral culture, antigen and antibody detection assays and nucleic acid detection methods are amongst the diagnostics available for VHF agents with varying sensitivities and specificities. Viral culture remains the gold standard for diagnosis, but it is limited by its non-availability in resource-limited settings. Antigen and antibody detection assays has issues with their sensitivity and specificity particularly once there is resolution of the antigenemia period in majority of patients [4, 10]. The only available diagnostic method in most setting in resource-limited parts of endemic regions is the nucleic acid-based assays such as the Reverse Transcriptase Polymerase Chain Reaction (RT-PCR) [4, 10].

Viral hepatitis remains a major public health burden globally, driven by five unrelated hepatotropic viruses; hepatitis A virus (HAV), the hepatitis B virus (HBV), the hepatitis C virus (HCV), the hepatitis D virus (HDV), and the hepatitis E virus (HEV) [12, 13]. These hepatitis agents have been documented to be associated with high disease burden, morbidity, and mortality. HBV and HCV are responsible for about 90% of these mortalities, whilst the remaining 10% of mortalities are caused by other hepatitis viruses [13–15]. Hepatitis A and E viruses cause hepatitis in both humans and various mammals and affect the liver like other hepatitis viruses [16–18]. Hepatitis A and E viruses are typically transmitted faecal-orally by ingestion of contaminated food or water [19]. In most cases, hepatitis A and E infections are asymptomatic or could be self-limiting with complete resolution within a few weeks without treatment [20–23]. Globally, an estimated 101.7 and 28.4 million cases of HAV and HEV respectively have been documented as a result of sporadic cases and outbreaks reported mostly in resource-limited settings [24]. In 2015, approximately 20 million people were reported to be infected by HEV with 44,000 deaths by the World Health Organization (WHO) [25]. These feacal-orally transmitted hepatitis agents have major public health implications in sub-Saharan Africa as HEV outbreaks is reported nearly annually with some outbreaks involving more than 10,000 cases [12, 16, 26]. In Nigeria, there have been documented serologic evidence of previous exposure to HAV infection in most children by five years of age [5, 12, 27, 28]. However,

the actual burden and prevalences of both HAV and HEV are grossly underestimated in Africa due to some deficiencies in the surveillance and monitoring systems [29].

Transmission of Hepatitis B, C and D viruses majorly occurs via contact with infected blood, blood products and other body fluids such as semen, vaginal fluids, and saliva [30]. Therefore, communal modes of spread are by transfusion of unscreened blood and its products, unprotected sexual activities, mother-to-child at birth, use of contaminated or inadequately sterilized instruments and sharing of sharp objects particularly during some traditional or cultural practices [31–33]. Individuals infected with these set of hepatitis viruses could develop chronic hepatitis, liver cirrhosis, and hepatocellular carcinoma with high morbidity and mortality rates if there are no interventions [15, 30]. Globally, greater than 350 million people are chronically infected with HBV, while about 150 and 15 million are infected with HCV and HDV respectively [12, 34, 35]. Among those infected with HBV and HCV, sub-Saharan Africa remains a hotspot for these hepatitis viruses, and Nigeria is one of the most endemic countries in this region. The prevalence rates of both HBV and HCV ranges from 3 to 20% and 1 to 7% respectively within the general population [12, 36].

Nigeria is endemic for both viral hepatitis and haemorrhagic fevers, but the actual interactions and disease burden of co-infections of these agents is yet to be unravelled in the country. Over the decades, Nigeria has experienced yearly outbreaks of VHFs especially Lassa fever, with an increasing trend of morbidity and mortality in a significant number of states since 2016 [10, 37]. A surge in the number of suspected Lassa fever cases was documented in Nigeria, increasing from 273 cases in 2016 to 1,022 in 2017, 3,498 in 2018, and 3,728 in 2019, with most of these cases originating from Edo and Ondo states [2, 38–40]. A similar trend of the increase in the case density and geographic distribution of Lassa fever across the states in Nigeria from 2018 to 2020 has also being documented by Dalhat et al., 2022 [41]. Despite the increasing trends of Lassa fever suspected cases in Nigeria, Chikungunya and West Nile viruses have also been detected in febrile patients in Ile-Ife, Osun state [42], while Dengue was reported from patients in Ibadan [43] and Ilorin [44]. However, amongst the VHF-suspected cases in Nigeria, only a few cases are laboratory-confirmed with the majority of cases unconfirmed for any aetiologic agent in Nigeria. Over the last two decades at the Centre for Human and Zoonotic Virology (CHAZVY), College of Medicine of the University of Lagos (CMUL), approximately 93.5% of suspected cases of VHF had been confirmed negative for Lassa, Yellow fever and Dengue viruses when tested in our facility. Therefore, this study aimed to investigate the detection of hepatitis viruses (HAV, HBV, HCV and HEV) as co-morbidities and evaluate the different patterns of co-infections in suspected but unconfirmed cases of viral haemorrhagic fever in Nigeria.

## Materials and methods

### Study design/location/population

This was a retrospective cross-sectional study carried out at the Centre for Human and Zoonotic Virology (CHAZVY), College of Medicine of the University of Lagos (CMUL), Idi-Araba, Lagos, Nigeria between January, and March 2022. A total of three hundred and fifty (350) samples from outbreaks and sporadic cases confirmed negative by Reverse Transcriptase Polymerase Chain Reaction (RT-PCR) for Lassa virus, Yellow fever, and Dengue virus respectively from individuals aged between <1 and 70 years were analyzed in this study. These samples were retrieved from the -80˚C sample bank facility at CHAZVY of suspected cases of VHF samples received between 2017 and 2020 from Lassa fever treatment centers across the country.

## Ethical consideration

Ethical approval for this study was received from the Health Research Ethics Committee (HREC), College of Medicine of the University of Lagos (CMUL) with the approval number: CMUL/HREC/02/20/718. Fully anonymized confidential data were accessed, kept confidential, stored in access-protected computer system, and analyzed as part of VHF public health response in Nigeria. The samples were stored at the Centre for Human and Zoonotic Virology (CHAZVY) and accessed with permission from the laboratory management.

## Data/specimen retrieval and storage

Anonymized laboratory unique codes for approximately 20% of all the negative plasma samples of suspected VHF cases received from Lassa treatment facilities within the country and stored at CHAZVY were randomly selected by the data manager. Corresponding plasma samples of the selected codes were sorted from the -80˚C storage freezers using the Universal precautions as recommended by the Centre for Diseases Control and Prevention, between Monday 8th to Friday 12th June 2020. [45, 46]. The cold chain of the samples was maintained, and randomly selected plasma samples were returned to the -80˚C freezer until analysis.

## Serological analysis using Enzyme Linked Immunosorbent Assay (ELISA)

All selected stored plasma samples were analyzed and interpreted according to manufacturer's instructions (Aria, CTK Biotech Inc, USA) for the detection of antibodies for Hepatitis A Virus Immunoglobulin M (HAV IgM), Hepatitis C Virus Antibody (HCV Ab) and Hepatitis E Virus Immunoglobulin M (HEV IgM) and the detection of antigen for Hepatitis B Virus Surface Antigen (HBsAg) serologically.

## Nucleic acid extraction and Polymerase Chain Reaction (PCR)

Viral genetic materials from selected sample aliquots (140 μL) were extracted according to manufacturer's instructions in a biosafety level 2 laboratory using a mini spin column RNA and DNA extraction kits (Qiagen, Germantown, Maryland, United States) respectively. The extracted genetic materials (DNA/RNA) samples were used for Reverse Transcription PCR (RT-PCR) and PCR amplification respectively using the primer sets highlighted in Table 1. For HAV, HCV and HEV, 5 μL of the extracted RNA was amplified in 20 μL of RT-PCR master mix reagents (JENA HOT START MIX) prepared using 5 μL of master mix buffer, 1μL each of forward and reverse primers (Table 1), and 13 μL of Nuclease-free water. The amplification was done using the Applied Biosystems Thermocycler with the following cycling conditions; 30 mins at 50˚C, 5 mins at 95˚C, followed by 35 cycles of 15 secs at 95˚C, 15 secs at 55˚C, 20 secs at 68˚C and 3 mins at 68˚C for final elongation. For HBV, 3 μL of the extracted DNA was amplified with 22 μL master mix prepared using 5 μL of master mix buffer, 1 μL each of forward and reverse primers (Table 1) and 15 μL of Nuclease-free water. The PCR cycling conditions were 94˚C for 5 mins followed by 40 cycles of 30 secs at 94˚C, 30 secs at 55˚C, 20 secs at 72˚C and final elongation at 72˚C for 5 mins. The generated PCR amplicons were analyzed using 2.5% agarose gel electrophoresis with 1X SYBR® safe DNA gel staining dye (Invitrogen, Carlsbad, California, United States) run at 120V/400mA for 30 minutes. The resolved amplicon bands images were captured with the BioDocAnalyze 2.0 (Biometra, Goettingen, Germany) under UV light and stored on an access-protected computer system within CHAZVY.

**Table 1. Primer sequences for HAV, HBV, HCV and HEV Reverse Transcription (RT-PCR) and PCR amplification.**

| Virus | Primer | Sequences | Amplicon size (bp) | Reference |
|---|---|---|---|---|
| Hepatitis A Virus | HAVC-R, 5' | CTCCAGAATCATCTCCAAC-3' | 192bp | Tsai et al., 1994 |
| | HAVC-L, 5' | CAGCACATCAGAAAGGTGAG-3' | | |
| Hepatitis B Virus | P2f | 5'-CCT GCT GGT GGC TCC AGT TC (20)-3' | 1000+bp | Iris et al., 2012 |
| | 979 | 5'-CAA AAG ACC CAC AAT TCT TTG ACA TAC TTT CCA AT (35)-3' | | |
| | Mc2r_(rv) | 5'-TGG AAG TTG GGG ATC ATT GC (20)-3' | | |
| Hepatitis C Virus | HCV_A1 | 5'-GAT GCA CGG TCT ACG AGA CCT C (22)-3' | 200bp | Menha et al., 2011 |
| | HCV_S1 | 5'-AAC TAC TGT CTT CAC CCA GAA (21)-3' | | |
| | HCV_A2 | 5'-GCG ACC CAA CAC TAC TCG GCT (21)-3' | | |
| | HCV_S2 | 5'-ATG GCG TTA GTA TGA GTG (18)-3' | | |
| Hepatitis E Virus | ORF1F1 strand | 5'-CCA YCA GTT YAT HAA GGC TCC- 3' (21mer) | 170bp | Guiseppina et al., 2014 |
| | ORF1R1 strand | 5'-TAC CAV CGC TGR ACR TC-3' (17mer) | | |
| | ORF1Fnstd | 5'-CTC CTG TTG GCR TYA CWA CTE C- 3' (19mer) | | |
| | ORF1Rnstd | 5'- GGR TGR TTC CAI ARV ACY TC- 3 (20mer) | | |

## Statistical analysis

All data were entered in the computer and analysed using SPSS version 26.0 for Windows (IBM Corp, 2021). Descriptive statistics were computed for all relevant data. The association between different demographics and the outcome variables was tested using Chi-square. All significant associations were recorded at $p \leq 0.05$.

## Results

A total of 350 retrieved samples from suspected VHF cases collected between 2017 and 2020 at CHAZVY were screened for HAV and HEV IgM, HBsAg and anti-HCV by ELISA. The participants included more males, 212 (60.6%) than the females, 138 (39.4%) and majority (92.0%) are alive, while only 7 (2%) of those who are alive are pregnant (Table 2). Their age ranged from <1–70 years, and the mean age of all participants was 25 ±14.5 years. The age range with the highest number of participants was 21–30 years with 98 (28.0%) participants, followed by 11–20 years with 71 (20.3%) and the range with the least number of participants is ≤1 with 4 (1.1%) (Table 2). Most of these suspected cases samples were sent from the South-west, North-east and North-west of Nigeria with frequency rates of 35.1%, 34.3% and 15.4% respectively and the most documented symptoms were loss of appetite (88.6%), general weakness (88%), fever (87.4%), joint pain (80.6%) and Jaundice (76.6%) with 12.6% having both fever and bleeding (Table 2). 37.7% of the participants had no formal education, while 28%, 24% and 10% had primary, secondary and tertiary education respectively (Table 2).

Out of the 350 samples analyzed, 126 (36%) of these VHF suspected cases were positive for at least one of the hepatitis viruses by ELISA (Table 3). Among the 126 (36%) VHF suspected cases that were positive for hepatitis viruses, the distribution of the specific hepatitis virus (single infection) detected revealed that none (0%) was reactive for HAV, while 37 (10.6%), 18 (5.1%) and 71 (20.3%) were positive for HBV, HCV and HEV respectively (Table 3). Further evaluation of co-infections of hepatitis viruses in these cohorts of positive individuals showed an overall co-infection rate of 15 (11.9%). Dual co-infections of HBV/HCV, HBV/HEV, HCV/HEV and triple co-infections of HBV/HCV/HEV were observed with prevalence rates of 3 (2.4%), 2 (1.6%), 8 (6.3%) and 2 (1.6%) respectively (Table 3). Majority (62.7%) of those positive for at least one hepatitis virus were males, compared to their female counterparts who had

**Table 2. Socio-demographics characteristics of study participants (n = 350).**

| PARAMETER | LEVEL | FREQUENCY (%) |
|---|---|---|
| **GENDER** | MALE | 212 (60.6) |
| | FEMALE | 138 (39.4) |
| **AGE** | <1 | 4 (1.1) |
| | 1–10 | 65 (18.6) |
| | 11–20 | 71 (20.3) |
| | 21–30 | 98 (28.0) |
| | 31–40 | 66 (18.9) |
| | 41–50 | 31 (8.9) |
| | 51–60 | 9 (2.6) |
| | 61–70 | 6 (1.7) |
| **GEOGRAPHIC ZONE** | NORTH-CENTRAL | 26 (7.4) |
| | NORTH-EAST | 120 (34.3) |
| | NORTH-WEST | 54 (15.4) |
| | SOUTH-EAST | 1 (0.3) |
| | SOUTH-SOUTH | 26 (7.4) |
| | SOUTH-WEST | 123 (35.1) |
| **SYMPTOMS** | FEVER | 306 (87.4) |
| | FEVER/BLEEDING | 44 (12.6) |
| | VOMITING/NAUSEA | 236 (67.4) |
| | DIARRHEA | 194 (55.4) |
| | GENERAL WEAKNESS | 308 (88.0) |
| | LOSS OF APPETITE | 310 (88.6) |
| | ABDOMINAL PAIN | 221 (63.1) |
| | JOINT PAIN | 282 (80.6) |
| | JAUNDICE | 268 (76.6) |
| **EDUCATION LEVEL** | NO FORMAL | 132 (37.7) |
| | PRIMARY | 98 (28.0) |
| | SECONDARY | 84 (24.0) |
| | TERTIARY | 36 (10.3) |
| **LIVE STATUS** | DEAD | 28 (8.0) |
| | ALIVE | 322 (92.0) |
| | ALIVE & PREGNANT | 7 (2.0) |

a prevalence of 37.3%. Most of the positive hepatitis samples were within age groups 21–30, 11–20 and 31–40 years with prevalence rates of 36 (10.3%), 31 (8.9%) and 21 (6%) respectively (Table 3). The positivity rates of at least one hepatitis virus detected based on the geographic regions of Nigeria showed that the North-East region had the highest number of positives with 67 (19.1%), the North-West region with 23 (6.6%) and the South-West region with 22 (6.2%) positives. The South-South and North-Central both had 7 (2%) positive rates respectively. However, no positive case was detected among samples from South-East region (Table 3).

Comparative analysis of the distribution of hepatitis positive samples of the geographic regions with dual and triple co-infections showed that out of 15 positive samples with evidence of co-infection, majority (40%) were detected from the North-East region, with an even distribution (50%) between the males and females (Table 4). Twenty percent (20%) each were positive from the North-West, South-West and South-South regions respectively, while there was no (0%) co-infection of hepatitis viruses among suspected cases from the North-Central and South-East regions (Table 4). Similar male to female positive ratios of 2 to 1 were observed in

**Table 3. Distribution of hepatitis A, B, C and E positivity among the participants and their geographic location (n = 350).**

| VARIABLES | POSITIVE (%) | P values |
|---|---|---|
| **Hepatitis Status** | | |
| HAV | 0 (0.0) | |
| HBV | 37 (10.6) | |
| HCV | 18 (5.1) | |
| HEV | 71 (20.3) | 0.025 |
| **Total** | 126 (36) | |
| **Hepatitis Co-infection** | | |
| HBV/HCV | 3 (20.0) | |
| HBV/HEV | 2 (13.3) | |
| HCV/HEV | 8 (53.3) | |
| HBV/HCV/HEV | 2 (13.3) | 0.274 |
| **Total** | 15 (11.0) | |
| **Gender** | | |
| MALE | 79 (62.7) | |
| FEMALE | 47 (37.3) | 0.230 |
| **Age Group** | | |
| <1 | 0 (0.0) | |
| 1–10 | 15 (4.3) | |
| 11–20 | 31 (8.9) | |
| 21–30 | 36 (10.3) | |
| 31–40 | 21 (6.0) | |
| 41–50 | 16 (4.6) | |
| 51–60 | 5 (1.4) | |
| 61–70 | 2 (0.6) | 0.198 |
| **Location** | | |
| NORTH-EAST | 67 (19.1) | |
| NORTH-WEST | 23 (6.6) | |
| NORTH-CENTRAL | 7 (26.9) | |
| SOUTH-WEST | 22 (6.2) | |
| SOUTH-SOUTH | 7 (2.0) | |
| SOUTH-EAST | 0 (0.0) | 0.242 |

the North-West and South-West regions, while a reverse ratio of male to female positive ratio of 1 to 2 was documented for the South-South region (Table 4).

The reverse transcriptase polymerase chain reaction (RT-PCR) investigation carried out on the 126 samples which tested positive for at least one hepatitis virus by Enzyme Immunoassay revealed the presence of expected amplicon band sizes base pairs (bp) as confirmation for HBV (860bp), HCV (187bp) and HEV (171bp) respectively as highlighted in Figs 1–3. The expected amplicon band sizes for HBV, HCV and HEV genetic materials were as compared with their positive control bands as highlighted on the gel image (Figs 1–3). The negative control lane had no visible bands (Figs 1–3). However, no amplicon band was observed for the expected band sizes (192 bp) for HAV (Fig 4).

The characteristics of the 126 (36%) VHF suspected cases who tested positive for at least one hepatitis virus by Enzyme Immunoassay with a frequency positive rate of 37 (10.6%), 18 (5.1%) and 71 (20.3%) for HBV, HCV and HEV respectively with their RT-PCR results

**Table 4. Hepatitis co-infection distribution in geographic regions and gender.**

| REGIONS | No. of Samples (%) | | Total (%) | No. of +ve Samples (%) | | Total |
|---|---|---|---|---|---|---|
| | Male | Female | | Male | Female | |
| N/EAST | 63 (29.7) | 57 (41.3) | 120 (34.3) | 3 (37.5) | 3 (42.9) | **6 (40%)** |
| N/WEST | 38 (17.9) | 16 (11.6) | 54 (15.4) | 2 (25) | 1 (14.3) | **3 (20%)** |
| N/CENTRAL | 12 (5.7) | 14 (10.1) | 26 (7.4) | - | - | - |
| S/WEST | 85(40.1) | 38(27.5) | 123 (35.1) | 2(25) | 1(14.3) | **3 (20%)** |
| S/SOUTH | 14(6.6) | 12(8.7) | 26 (7.4) | 1(12.5) | 2(28.6) | **3 (20%)** |
| S/EAST | - | 1(0.7) | 1 (0.3) | - | - | - |
| TOTAL | **212 (60.6)** | **138 (39.4)** | **350 (100)** | **8 (53.3)** | **7 (46.7)** | **15 (100)** |
| P Value | | | | **0.242** | **0.285** | |

revealed that the male participants, Northeast and the most active age groups of 11–20, 21–30 and 31–40 years had higher infection rates for the three hepatitis viruses without statistically significant differences (Table 5). There were statistically significant differences between the rates of positive cases among samples from the geographic zones (P– 0.025) and the levels of education (P– 025) of these positive participants (Table 5). All (100%) of the 37 and 18 positive participants for both HBV and HCV and 68 (95.8%) of the 71 participants positive for HEV were still alive. One (1.4%) and 2 (2.8%) among those positive for HEV were documented to be dead and pregnant respectively with no statistically significant difference (P– 0.200) (Table 5). The RT-PCR results showed that 26 (70.3%), 12 (66.7%) and 59 (83.1%) of seropositive individuals had detectable DNA for HBV and detectable RNA for HCV and HEV respectively (Table 5).

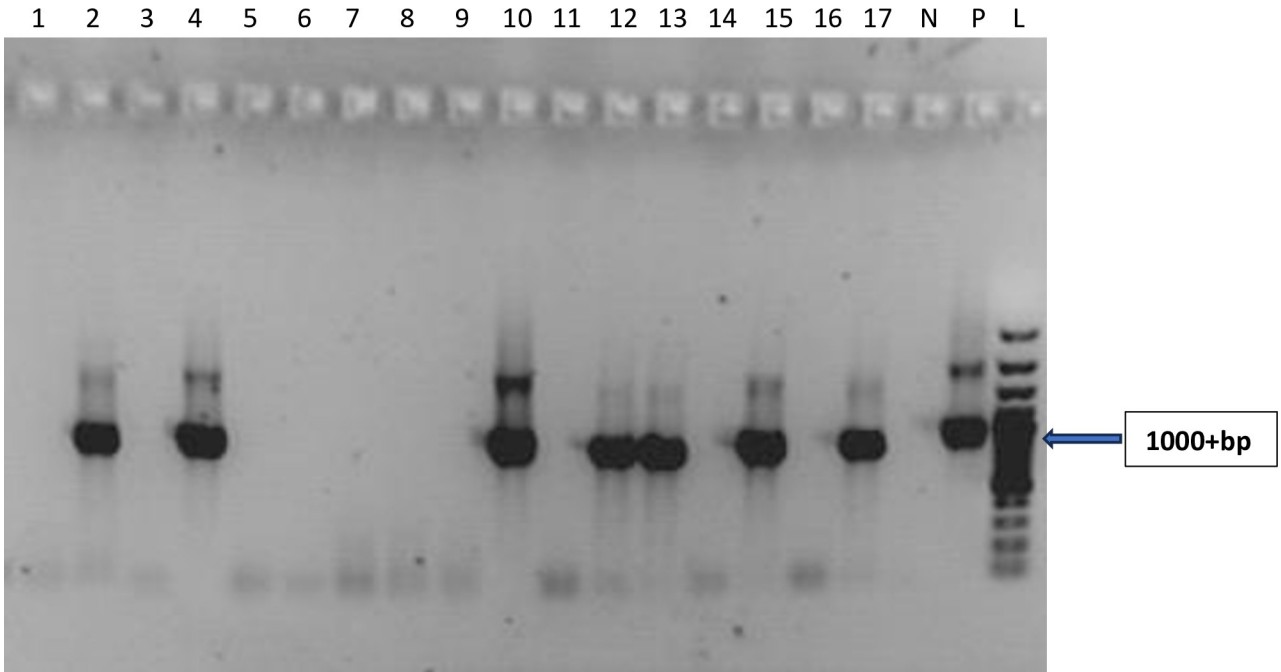

**Fig 1. Agarose gel image showing the PCR amplicon bands for hepatitis B virus DNA.** Samples on lanes 2, 4, 10, 12, 13, 15 and 17 were positive with an amplicon size of 1000+bp as compared with the positive control. N–Negative Control; P–Positive Control and L–DNA ladder (100bp).

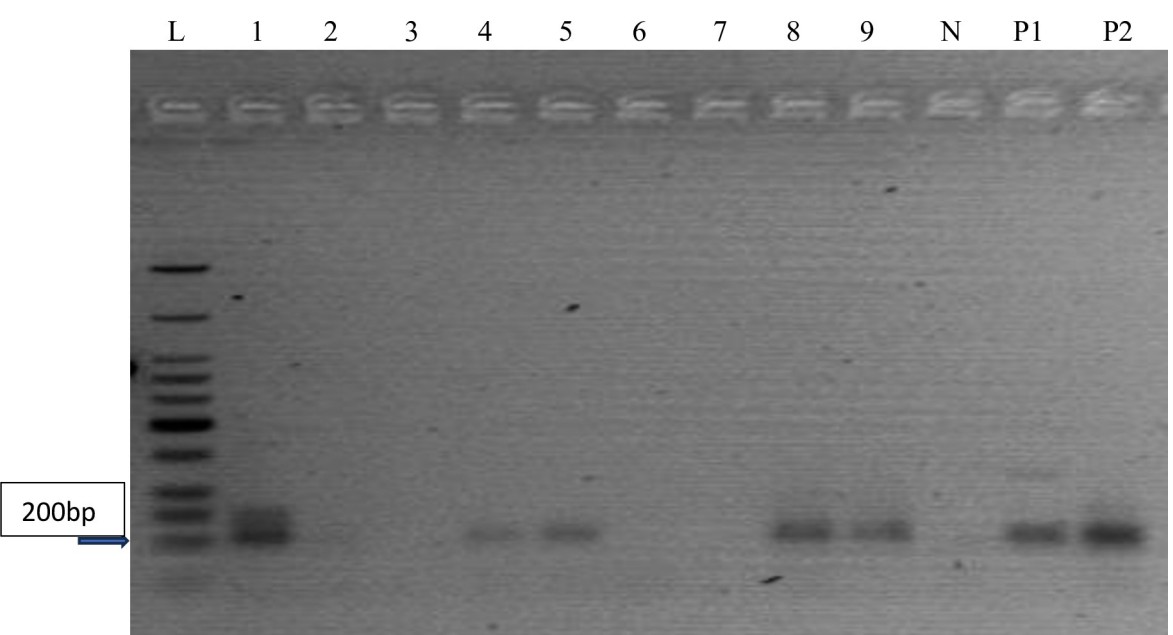

**Fig 2. Agarose gel image showing the PCR amplicon bands for hepatitis C virus RNA.** Samples on lanes 1, 4, 5, 8 and 9 were positive with an amplicon size of 200bp as compared with the positive control. N–Negative Control; P1 & P2 –Positive Control and L–DNA ladder (100bp).

## Discussion

The resource-limited countries, particularly in sub-Saharan Africa remain a beehive of febrile illnesses with multiple etiological agents which are difficult to distinguish based on the similarities of their signs and symptoms. The manifestation of febrile illnesses could be because of parasitic diseases such as Malaria, Toxoplasmosis, Schistosomiasis; bacteria diseases such as

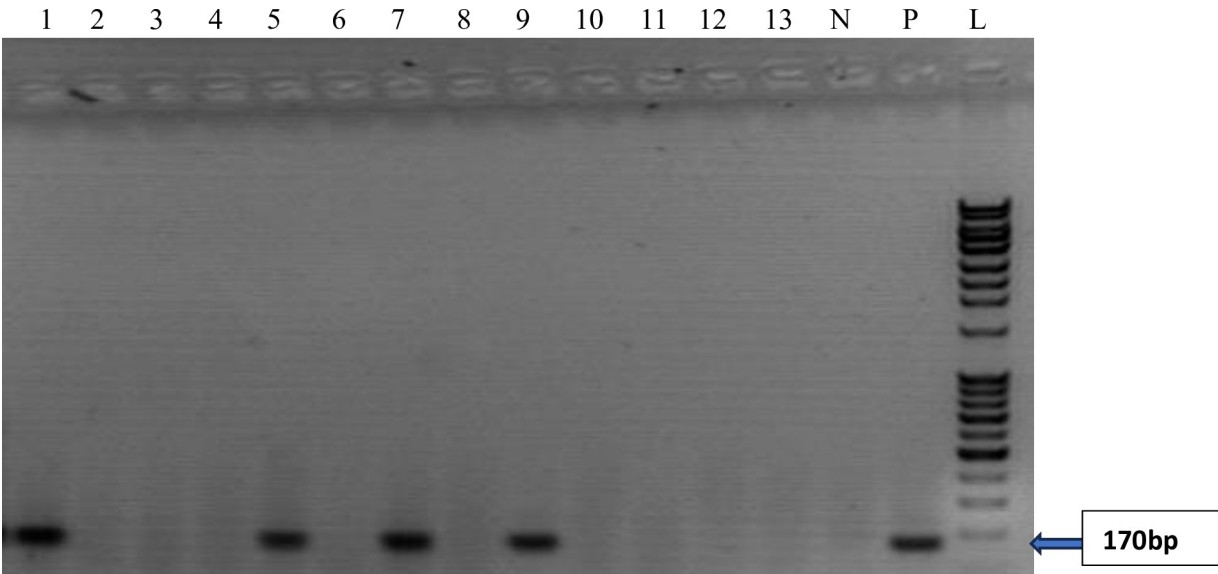

**Fig 3. Agarose gel image showing the PCR amplicon bands for hepatitis E virus RNA.** Samples on lanes 1,5,7 and 9 were positive with an amplicon size of 170bp as compared with the positive control. N–Negative Control; P–Positive Control and L–DNA ladder (100bp).

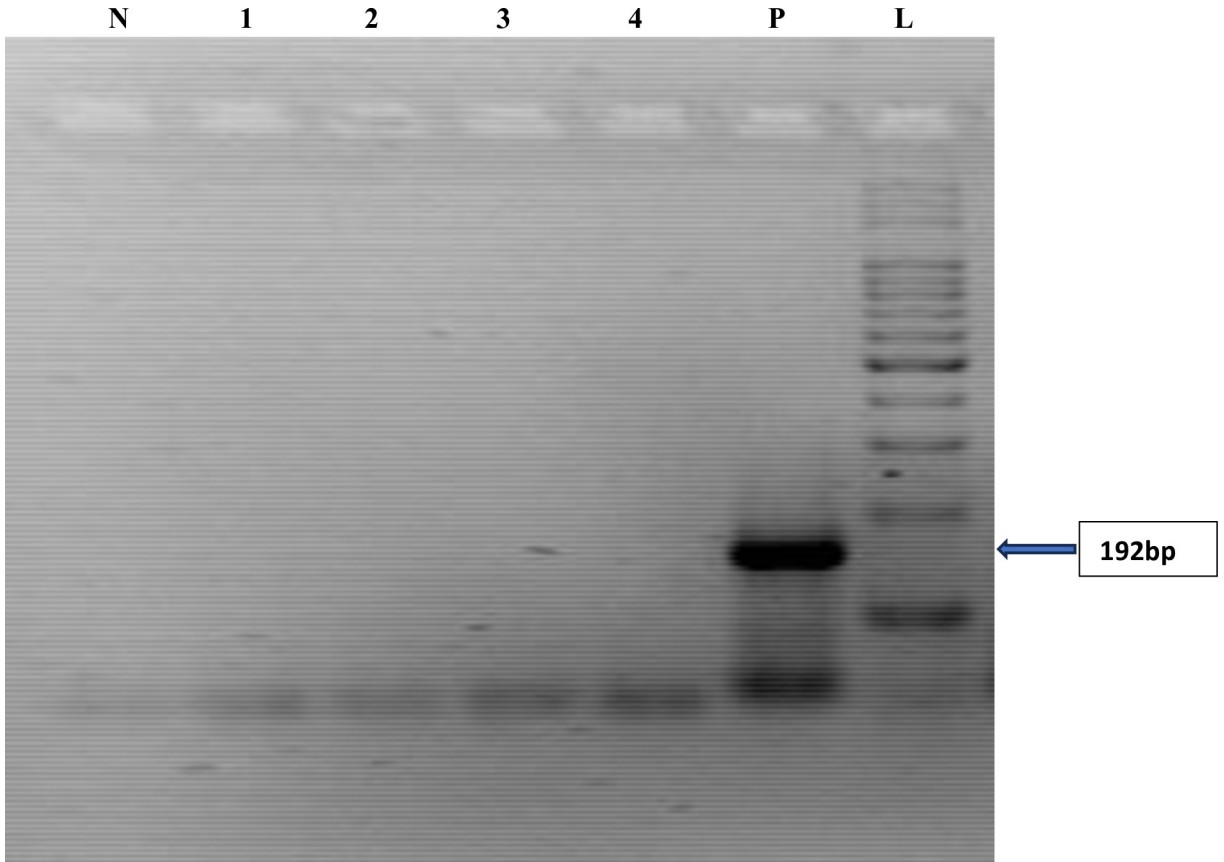

**Fig 4. Agarose gel image showing the PCR amplicon bands for hepatitis A virus RNA.** Samples on lanes 1, 2, 3 and 4 were negative as compared with the positive control (192bp). N–Negative Control; P–Positive Control and L–DNA ladder (100bp).

Typhoid, Typhus, Borreliosis, Leptospirosis; or by viral diseases such as Cytomegalovirus, Mumps, Measles, Rubella, Coxsackie, Yellow fever, Hepatitis, Lassa, Marburg, Ebola, Dengue, Crimean–Congo and Hantaan viral haemorrhagic fevers [3, 5, 8, 9, 28, 47–49]. Over the decades, these diseases have been major public health problems in the sub-Saharan African region with limited or no information on the actual burdens of these agents and their interactions as co-morbidities and mortalities in febrile illnesses. A better understanding of the in-country epidemiology of hepatitis viruses and other febrile illnesses' agents could guide informed policy for their possible inclusion as differential diagnoses in suspected cases of VHF [5, 47]. Nigeria had witnessed a surge in the number of states reporting suspected cases of VHF since 2016 [10, 41]. Viral hepatitis and VHF infections are two public health problems with endemicity in-country and there are difficulties in distinguishing VHFs from other endemic febrile agents circulating in Nigeria and other African countries. Considering that only limited number of VHF agents are routinely tested for in Nigeria, there is dearth of unswerving epidemiological data on the contributions of other aetiologic agents that present with similar symptoms as VHFs. The preliminary findings of the detection of some hepatitis viruses in unconfirmed suspected cases of VHF requires further evaluation to determine the possible inclusion of these agents as differential causative agents and their contributions as co-morbidities and mortalities in suspected but unconfirmed cases of VHFs in Nigeria.

Findings from this study suggest that hepatitis virus infections might be considered as differential aetiologic agents that might be a major comorbidity in the outbreaks of viral

**Table 5. Demographic characteristics of serology positive hepatitis B, C and E and RT-PCR analysis among the participants (n = 126).**

| SOCIO-DEMOGRAPHICS | HBV POSITIVES (%) | HCV POSITIVES (%) | HEV POSITIVE (%) | P Values |
|---|---|---|---|---|
| | (N = 37) | (N = 18) | (N = 71) | |
| **GENDER** | | | | |
| MALE | 26 (70.3) | 11 (61.1) | 45 (63.4) | 0.950 |
| FEMALE | 11 (29.7) | 7 (38.9) | 26 (36.6) | |
| **AGE GROUP** | | | | |
| <1 | 0 (0) | 0 (0) | 0 (0) | 0.950 |
| 1–10 | 3 (8.1) | 0 (0) | 12 (16.9) | |
| 11–20 | 8 (21.6) | 2 (11.1) | 21 (29.6) | |
| 21–30 | 10 (27.0) | 6 (33.3) | 20 (28.2) | |
| 31–40 | 7 (18.9) | 5 (27.8) | 9 (12.7) | |
| 41–50 | 7 (18.9) | 3 (16.7) | 6 (8.5) | |
| 51–60 | 1 (2.7) | 1 (5.6) | 3 (4.2) | |
| 61–70 | 1 (2.7) | 1 (5.6) | 0 (0) | |
| **GEOGRAPHIC ZONE** | | | | |
| NORTH-CENTRAL | 5 (13.5) | 1 (5.6) | 1 (1.4) | 0.025 |
| NORTH-EAST | 18 (48.6) | 9 (50.0) | 40 (56.3) | |
| NORTH-WEST | 6 (16.2) | 3 (16.7) | 14 (19.7) | |
| SOUTH-EAST | 0 (0) | 0 (0) | 0 (0) | |
| SOUTH-SOUTH | 4 (10.8) | 3 (16.7) | 0 (0) | |
| SOUTH-WEST | 4 (10.8) | 2 (11.1) | 16 (22.5) | |
| **SYMPTOMS** | | | | |
| FEVER | 38 (100) | 18 (100) | 71 (100) | 0.274 |
| FEVER/BLEEDING | 3 (8.1) | 0 (0) | 0 (0) | |
| VOMITING/NAUSEA | 28 (73.7) | 13 (72.2) | 69 (97.2) | |
| DIARRHEA | 18 (47.4) | 13 (72.2) | 68 (95.8) | |
| GENERAL WEAKNESS | 34 (89.5) | 17 (94.4) | 60 (84.5) | |
| LOSS OF APPETITE | 35 (92.1) | 14 (77.8) | 55 (77.5) | |
| ABDOMINAL PAIN | 28 (73.7) | 14 (77.8) | 68 (95.8) | |
| JOINT PAIN | 28 (73.7) | 13 (72.2) | 38 (53.5) | |
| JAUNDICE | 34 (89.5) | 11 (61.1) | 36 (50.7) | |
| **EDUCATION LEVEL** | | | | |
| NO FORMAL | 12 (32.4) | 5 (27.8) | 28 (39.4) | 0.025 |
| PRIMARY | 9 (24.3) | 5 (27.8) | 25 (35.2) | |
| SECONDARY | 6 (16.2) | 5 (27.8) | 16 (22.5) | |
| TERTIARY | 10 (27.0) | 3 (11.7) | 2 (2.8) | |
| **LIVE STATUS** | | | | |
| DEAD | 0 (0) | 0 (0) | 1 (1.4) | 0.200 |
| ALIVE | 37 (100) | 18 (100) | 68 (95.8) | |
| ALIVE & PREGNANT | 0 (0) | 0 (0) | 2 (2.8) | |
| **RT-PCR STATUS** | | | | |
| POSITIVE | 26 (70.3) | 12 (66.7) | 59 (83.1) | 0.230 |
| NEGATIVE | 11 (29.7) | 6 (33.3) | 12 (16.9) | |

haemorrhagic fevers in Nigeria, as 36% of these samples suspected of VHF but confirmed to be negative for Lassa, Yellow fever and Dengue viruses were reactive for at least one hepatitis virus. The association of these hepatitis viruses with other VHF agents might be crucial in the differential diagnosis and management of these endemic diseases within our environment.

These agents have been documented to possess similar clinical symptoms such as renal failure, hepatic injury, jaundice, fever, fatigue, loss of appetite, nausea, or vomiting [50–52]. Thus, the effects of these viral hepatitis agents on the burden of VHFs in Nigeria cannot be ruled out and as such it is methodical to diagnose other viral hepatitis to determine its effect on VHFs on a routine basis [1, 48]. Several studies have documented co-infections of some hepatitis viruses in suspected cases of VHFs in some parts of Africa such as HAV and HCV co-infections as well as in Dengue and yellow fever infections but not in Nigeria [5, 28, 47, 48, 53, 54]. This present study therefore provides documented evidence of the detection of hepatitis viruses in the cohort of suspected but unconfirmed cases of VHFs in Nigeria. Co-infections of these hepatitis viruses, particularly in endemic countries, should be further investigated since the pathway of liver involvement and degeneration in VHF cases has not been fully elucidated [28, 54]. In endemic areas, more studies to ascertain the actual burden, co-circulation and clinical implications of hepatitis and other viruses with febrile manifestations as VHFs is advocated in Nigeria. Once established, it might be reasonable to screen for these viruses in suspected cases of VHFs since signs and symptoms are superimposed on each other which could lead to missed diagnosis [1, 3, 5].

Based on the individual hepatitis virus detected among these suspected cases of VHFs in Nigeria, data from this study revealed HEV infections as the most prevalent (20.3%), followed by HBV and HCV with prevalence rates of 10.6% and 5.1% respectively. None was positive for HAV infection in this study population. These findings further confirmed the endemicity of these hepatitis viruses within our environment. However, it is not surprising that none of these individuals was positive for Hepatitis A Virus IgM in an endemic country such as Nigeria. This might be connected to the fact that, nearly all children might have encountered HAV at an early age with asymptomatic manifestations, expression of serological evidence of prior exposure and the development of long-term immunity [12, 33, 55–58]. The detection rates of HAV and HEV were not comparable in this study, despite that both viruses are transmitted through contaminated water and food sources [59, 60]. The detected HEV prevalence was statistically significant in this study and shows there is high activity of the virus in Nigeria. This suggests that the endemicity of this agent among suspected cases of VHFs in our environment requires extensive evaluations. Thus, HEV might be a major additional disease burden fuelling the increasing numbers of suspected cases of VHFs as reported from known endemic and non-endemic states recently in Nigeria [61–64]. The evidence of HEV as documented here was significantly higher than the 0.4% obtained among pregnant women in Oyo and Anambra communities [61], 0.9% among different populations in Plateau State [62], and the 2.7% in apparently healthy participants in Ogbomoso, South-western Nigeria [63]. However, the finding from this study was like the 17.8% seroprevalence rate obtained from a study in Lagos [64].

The detected HBV and HCV prevalence in this study was at a ratio of 2:1, showing that HBV was twice as likely to be more prevalent in this category of population in our environment. This data also confirms the endemicity and wide spread of both viruses in sub-Saharan Africa including Nigeria with common transmission routes. The highest prevalence rates of between 3–20% and 1–7% for both HBV and HCV respectively have also been documented to be domicile among the general population within the sub-Saharan Africa region [12, 36]. However, varied rates for both HBV and HCV prevalence have been documented in Nigerian population with an estimated 14% and 2.1% prevalence rates of Hepatitis B virus and Hepatitis C infections respectively [65–67]. The prevalence of HBV and HCV documented in this study population might have significant consequences on the management of suspected cases of VHF as both viruses remain silent killers within our environment. Co-infection patterns observed highlight that HCV/HEV co-infection was mostly prevalent which was even higher than that of HBV/HCV co-infection which was surprising but not statistically significant in

this study. However, risk factors such as blood transfusion, scarification, circumcision, and heterosexual activities that might be associated with these co-infections among these suspected cases that were negative for any VHF agents in Nigeria were not evaluated in this study. The lower co-infection rates of HBV with other hepatitis viruses in this study could be due to the increased public awareness programmes and improved vaccination uptakes reported within the country. More follow-ups regarding HBV awareness have been done with routine investigations carried out during blood screenings and even in pregnant women. Also, business owners and industries including government establishments have mandated HBV vaccination for their staff.

The Northern region, especially the North-East recorded the highest prevalence of both single and hepatitis virus co-infection amongst this cohort of suspected cases of VHF in Nigeria particularly for HEV infections with statistically significant differences. This finding from the North-East might not be unconnected to various risk factors such as level of education, insecurity, vulnerability of children, level of hygiene, level of contamination, overpopulation, poor electricity supply, and inadequate water supply [58, 62, 67, 68]. Further evaluation of the drivers of these infections in this part of the country requires further investigation. The highest frequencies of at least a single hepatitis infection and co-infections were documented amongst age groups 11–20 and 21–30 years respectively in this study population. These are a very active group and most of them want unhindered freedom. They engage in lots of activities which could be social or environmental, some of them even care about their social status and outward appearance rather than their health status. At this age, individuals feel caged and controlled and therefore want their freedom. They get involved in acts and practices that could predispose to hepatitis infections such as drug abuse, drawing tattoos underneath the skin, and pre-marital sex amongst others. The high HEV prevalence in these age groups were comparable with other studies of HEV infections in Ogbomosho and Ibadan in Nigeria [63, 69], but were at variance with the low prevalence of HEV reported among community dwellers in Oyo and Anambra by Ifeorah *et al.,* [61] which all support the evidence of the endemicity of this agent in the country. This age group often eats out for the sake of convenience without caring about the hygiene levels of eateries that they patronize. Thus, they might be prone to ingestion of contaminated food and water. A higher prevalence of HEV in males as compared to females as seen in this study was comparable to the studies by Buti *et al.,* in 2008 [70]; and Oladipo *et al.,* in 2017 [63] but was at variance with the report by Fowotade *et al.,* in 2018 [69]. Generally, observations have shown that the female population has a better hygiene level compared to the male population, this also could be a reason for the higher numbers of affected males than females.

## Conclusion

The evidence of hepatitis virus infections in suspected cases of viral hemorrhagic fever in Nigeria as documented in this study suggests the inclusion of differential diagnosis for hepatitis viruses among this cohort of individuals. However, their associations as co-morbidities and mortalities in this category of individuals require further investigations in endemic countries such as Nigeria should be thoroughly evaluated to guide informed policy on the diagnosis and management of these cases.

## Supporting information

**S1 Raw images. Raw images of agarose gel documentation for hepatitis A, B, C and E viruses.**
(PDF)

## Acknowledgments

The authors acknowledge and thank the data officer of the Centre for Human and Zoonotic Virology (CHAZVY), College of Medicine, University of Lagos (CMUL) for assisting with the extraction of data.

## Author Contributions

**Conceptualization:** Olumuyiwa Babalola Salu, Roosevelt Amaobichukwu Anyanwu, Sunday Aremu Omilabu.

**Data curation:** Olumuyiwa Babalola Salu, Tomilola Feyikemi Akinbamiro, Remilekun Mercy Orenolu, Onyinye Dorothy Ishaya, Roosevelt Amaobichukwu Anyanwu, Olubunmi Rita Vitowanu, Maryam Abiodun Abdullah, Adenike Hellen Olowoyeye, Sodiq Olawale Tijani, Kolawole Solomon Oyedeji, Sunday Aremu Omilabu.

**Formal analysis:** Olumuyiwa Babalola Salu, Tomilola Feyikemi Akinbamiro, Remilekun Mercy Orenolu, Onyinye Dorothy Ishaya, Roosevelt Amaobichukwu Anyanwu, Maryam Abiodun Abdullah, Adenike Hellen Olowoyeye, Sodiq Olawale Tijani, Kolawole Solomon Oyedeji, Sunday Aremu Omilabu.

**Funding acquisition:** Olumuyiwa Babalola Salu, Sunday Aremu Omilabu.

**Investigation:** Olumuyiwa Babalola Salu, Sunday Aremu Omilabu.

**Methodology:** Olumuyiwa Babalola Salu, Tomilola Feyikemi Akinbamiro, Remilekun Mercy Orenolu, Onyinye Dorothy Ishaya, Roosevelt Amaobichukwu Anyanwu, Olubunmi Rita Vitowanu, Maryam Abiodun Abdullah, Adenike Hellen Olowoyeye, Sodiq Olawale Tijani, Kolawole Solomon Oyedeji, Sunday Aremu Omilabu.

**Project administration:** Olumuyiwa Babalola Salu, Kolawole Solomon Oyedeji, Sunday Aremu Omilabu.

**Resources:** Olumuyiwa Babalola Salu, Sunday Aremu Omilabu.

**Software:** Olumuyiwa Babalola Salu, Sunday Aremu Omilabu.

**Supervision:** Olumuyiwa Babalola Salu, Roosevelt Amaobichukwu Anyanwu, Sunday Aremu Omilabu.

**Validation:** Olumuyiwa Babalola Salu, Sunday Aremu Omilabu.

**Visualization:** Olumuyiwa Babalola Salu, Sunday Aremu Omilabu.

**Writing – original draft:** Olumuyiwa Babalola Salu, Tomilola Feyikemi Akinbamiro, Remilekun Mercy Orenolu, Onyinye Dorothy Ishaya, Kolawole Solomon Oyedeji, Sunday Aremu Omilabu.

**Writing – review & editing:** Olumuyiwa Babalola Salu, Tomilola Feyikemi Akinbamiro, Remilekun Mercy Orenolu, Onyinye Dorothy Ishaya, Roosevelt Amaobichukwu Anyanwu, Olubunmi Rita Vitowanu, Maryam Abiodun Abdullah, Adenike Hellen Olowoyeye, Sodiq Olawale Tijani, Kolawole Solomon Oyedeji, Sunday Aremu Omilabu.

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
