## [Decision Letter · Decision Letter 0]

19 Feb 2024

PONE-D-23-34199Detection of Hepatitis Viruses in Suspected Cases of Viral Haemorrhagic Fevers in NigeriaPLOS ONE

Dear Dr. Salu,

Thank you for submitting your manuscript to PLOS ONE. After careful consideration, we feel that it has merit but does not fully meet PLOS ONE’s publication criteria as it currently stands. Therefore, we invite you to submit a revised version of the manuscript that addresses the points raised during the review process.

**ACADEMIC**
**EDITOR**: Please, improve the text of the manuscript as it is presently clumsy. Also, provide a description of the outcome of molecular analyses. It would be better if authors carried out sequencing of amplified products. 

We look forward to receiving your revised manuscript.

Kind regards,

Daniel Oladimeji Oluwayelu, D.V.M., M.Sc., Ph.D.

Academic Editor

PLOS ONE

Journal Requirements:

Reviewers' comments:

Reviewer's Responses to Questions

**Comments to the Author**

1. Is the manuscript technically sound, and do the data support the conclusions?

Reviewer #1: No

Reviewer #2: Yes

Reviewer #3: Yes

2. Has the statistical analysis been performed appropriately and rigorously? 

Reviewer #1: N/A

Reviewer #2: Yes

Reviewer #3: Yes

3. Have the authors made all data underlying the findings in their manuscript fully available?

Reviewer #1: Yes

Reviewer #2: Yes

Reviewer #3: Yes

4. Is the manuscript presented in an intelligible fashion and written in standard English?

Reviewer #1: Yes

Reviewer #2: Yes

Reviewer #3: Yes

5. Review Comments to the Author

Reviewer #1: In this manuscript, the authors described the outcome of a serological survey and molecular analysis of 350 Nigerian patients initially suspected of being affected from a viral hemorrhagic fever. All samples turned out to be negative for Lassa, Yellow Fever, and Dengue. Samples of these patients coming from all Nigerian provinces were screened by ELISA for the presence of serological scars signaling an infection with hepatitis A, B, C, or E viruses. Samples positive in ELISA were then analyzed by nested PCR.

A subset of 36% of patients presented a reactive serology. The most frequent was anti-Hepatitis E virus (20.3%). The authors suggest that in a certain number of cases, it is not a Haemorrhagic Fever virus (HFV) that is involved in the suspected disease but rather a more common hepatitis virus. They suggest to systematically extend screening of negative HFV to hepatitis.

The paper is interesting but asks many additional questions left unanswered and present some shortcomings as well.

The identification of the different causes of febrile illness is the topic of many publications in Africa and is not at all a trivial issue (see for example M.J. Maze et alii, The epidemiology of febrile illness in sub-Saharan Africa: implications for diagnosis and management. Clinical Microbiology and Infection 24, 2018, 808-814)

What is the proportion of samples routinely considered as negative for LASV, YFV or DENV in the laboratory of the authors?

The choice of the serological tests is puzzling. The authors targeted acute hepatitis A by using IgM anti-HAV kit but they looked in the meantime for markers used to detect chronic diseases for HCV and HBV or for past disease for HEV. They are thus mixing interpretations for acute and chronic hepatitis. It is probable that all anti-HCV or HBsAg positive patients were chronically infected for a while and that the fever that motivated the haemorrhagic fever diagnosis was not due to HBV or HCV. The 5% and 10% of smaples positive for HCV and HBV correspond roughly to what can be expected in the general Nigerian population while these figures should have been higher if they have been associated with the current disease. In addition, looking for anti-HEV IgG is also looking for past infection. Why not using anti-HBc IgM or anti-HEV IgM to detect an acute infection.

Why not looking for other HFV such as Chikungunya or West Nile Virus involved at high frequency in a former Nigerian survey (see Baba M et alii. Evidence of arbovirus co-infection in suspected febrile malaria and typhoid patients in Nigeria. J Infect Dev Ctries 2013;7:51e9) ?

“The cold chain of the samples was maintained, and randomly selected plasma samples”: what was the system of selection? It can significantly impact the final outcome of the study for example if a large proportion of sample coming from the North-East region has been “selected”.

Please provide a table comparing the samples positive and negative for hepatitis viruses with the outcome of statistical tests.

Please provide a table describing each type of patients: HBsAg(+), anti-HCV(+), anti-HEV(+).

There is a lengthy paragraph about the co-infections that represent only 5% of the whole series. They are only 15, and it is not sure that relevant conclusions can be draw from such a small subset.

Were some of the samples part of outbreaks or were they only sporadic cases of suspected VHF?

“In endemic areas, the knowledge of these hepatitis and other viruses that might be cocirculating in suspected cases of VHFs »: the provided evidence do not support the co-circulation of hepatitis and VHF in Nigeria. Regarding the co-circulation of hepatitis viruses it is well documented by the publication of papers dedicated to co-infections.

Likewise “The study also shows that there might be possible co-morbidity between viral hemorrhagic fevers and hepatitis viruses” is not a realistic statement. The study is not showing that and it was not meant to do so.

“This finding supports the documented evidence that globally, over 350 and 150 million people are chronically infected with HBV and HCV respectively”: I do not think that the findings of the present manuscript support these figures.

We do not know anything about the percentage of positivity by PCR for any viruses.

Why analyzing HAV by PCR while it was all negative by serology?

We do not know whether the patients with bleeding or those who died were infected or not with hepatitis virus. We do not know their demographical status as well.

Discussion

The long paragraph about the North East region should be shorten and substantiated by more references.

“this leads to corrupt and immoral practices »: I do not think that we can deem a subset of the Nigerian youth as corrupted and immoral. For sure, some young people are susceptible to adopt practices dangerous for their own life.

Minor:

Abstract: Please indicate clearly that the 350 samples analyzed were negative for LASV, YFV, and DENV.

“126 were positive for at least one hepatitis virus”

Background: “These hepatitis agents has been documented to be associated with high disease burden, morbidity, mortality and has the potential to cause outbreaks and epidemics of about 50% of hepatitis cases worldwide”. What are the cause of the other 50% of hepatitis outbreak and epidemics? It is weird.

“selected sample aliquots were extracted » : What was the volume of plasma used for extraction?

Results: « 212 (60.6%) males and 138 (39.4%) females with a ratio of 1:1.5” Please, precise the order of sexes to calculate the ratio M:F or F:M ?

Smoking and drinking cannot be linked directly to viral hepatitis.

“The findings in these age groups agrees with other studies of HEV infections in Nigeria…”: What does it mean?

Table 1: Please provide the reference of the primers used.

Table 2: “geo-political”, “geographical” instead?

Reviewer #2: Detection of Hepatitis Viruses in Suspected Cases of Viral Haemorrhagic Fevers in Nigeria

The significance of this investigation cannot be overstated, particularly given the ongoing global climate changes and the prevalence of arboviruses and other co-morbidities. The Sub-Saharan Africa (SSA) region, grappling with numerous health challenges and inadequate healthcare infrastructure, stands to benefit greatly from the findings of this scientific study.

That being said, I would like to share a few observations and comments.

Comments:

Background: The literature review should encompass an examination of different diagnostic procedures, their respective limitations, as well as the challenges faced by low- and middle-income countries (LMICs) in terms of availability and adequacy of these diagnostic tools for identifying viral hemorrhagic fevers (VHF) and other coinfection and co-morbidities.

Method.

Comments:

A comprehensive table outlining the clinical data pertaining to the signs, symptoms, and specific regions of the participants should be incorporated within the methods section.

The population delineation by the author was insufficient. It is imperative for the authors to provide a comprehensive definition of this population, taking into account factors such as age group, pregnant and non-pregnant individuals, with particular attention to the implications on HEV.

Does the author thinks the coinfections, co-circulation, low or high prevalence could potentially be attributed to antibody dependence enhancement (ADE)? In the context of arboviruses, is it plausible that the vector (namely mosquitoes) might have been capable of carrying both viruses, or perhaps acquiring and transmitting them simultaneously?

Reviewer #3: The manuscript was generally well written. Because the manuscript was line numbered, it was difficult writing separate report so the corrections were made in the document. I think the authors should follow the format of the journal to make review process easier.

6. PLOS authors have the option to publish the peer review history of their article (what does this mean?). If published, this will include your full peer review and any attached files.

Reviewer #1: No

Reviewer #2: **Yes: **Peter Mac Asaga {MD, PhD)

Reviewer #3: No

---

## [Author Response · Author response to Decision Letter 0]

16 Apr 2024

A revised version of the manuscript that addresses the points raised during the review process is hereby submitted.

---

## [Decision Letter · Decision Letter 1]

27 May 2024

PONE-D-23-34199R1

Detection of Hepatitis Viruses in Suspected Cases of Viral Haemorrhagic Fevers in Nigeria PLOS ONE

Dear Dr. Salu,

Thank you for submitting your manuscript to PLOS ONE. After careful consideration, we feel that it has merit but does not fully meet PLOS ONE’s publication criteria as it currently stands. Therefore, we invite you to submit a revised version of the manuscript that addresses the points raised during the review process.

We look forward to receiving your revised manuscript.

Kind regards,

Daniel Oladimeji Oluwayelu, D.V.M., M.Sc., Ph.D.

Academic Editor

PLOS ONE

Journal Requirements:

Reviewers' comments:

Reviewer's Responses to Questions

**Comments to the Author**

1. If the authors have adequately addressed your comments raised in a previous round of review and you feel that this manuscript is now acceptable for publication, you may indicate that here to bypass the “Comments to the Author” section, enter your conflict of interest statement in the “Confidential to Editor” section, and submit your "Accept" recommendation.

Reviewer #4: (No Response)

Reviewer #5: All comments have been addressed

2. Is the manuscript technically sound, and do the data support the conclusions?

Reviewer #4: Yes

Reviewer #5: Yes

3. Has the statistical analysis been performed appropriately and rigorously? 

Reviewer #4: N/A

Reviewer #5: Yes

4. Have the authors made all data underlying the findings in their manuscript fully available?

Reviewer #4: Yes

Reviewer #5: Yes

5. Is the manuscript presented in an intelligible fashion and written in standard English?

Reviewer #4: Yes

Reviewer #5: Yes

6. Review Comments to the Author

Reviewer #4: The manuscript is well research and well written. However, the statistical methods used for the analysis was not stated, and there some grammatical errors that will need to be corrected; for example:

1. Page 6-Ethical considerations, line 4, "accessed protected" should be edited to "access-protected".

2. Page 7-Data/specimen retrieval and storage, line 3, "randomly by......" should be edited to "randomly selected by.....".

Reviewer #5: I have read the manuscript with keen interest. It is an article of interest in the field of virology.

1. The abstract informative is informative and reflect the body of the paper.

2. The introduction provided sufficient background information for readers in the immediate field to understand the problem/hypotheses and it ended with the objectives of the study.

3. The text was well arranged and the logic is clear.

4. There are no grammatical errors in this article.

5. The related concepts were clearly introduced and the readability is sufficient.

6. The proposed simulation /experiment /scheme is quite novel.

7. The theoretical analysis in this article is strong.

8. The reference section is informative and accurate.

I therefore, recommend the manuscript for your acceptance

7. PLOS authors have the option to publish the peer review history of their article (what does this mean?). If published, this will include your full peer review and any attached files.

Reviewer #4: **Yes: **Oche Ochai Agbaji

Reviewer #5: **Yes: **Iheanyi Omezuruike Okonko PhD

---

## [Author Response · Author response to Decision Letter 1]

29 May 2024

Responses has been made to the journal requirements and all reviewers comments and questions as requests.

---

## [Editor Report · Decision Letter 2]

3 Jun 2024

Detection of Hepatitis Viruses in Suspected Cases of Viral Haemorrhagic Fevers in Nigeria

PONE-D-23-34199R2

Dear Dr. Salu,

We’re pleased to inform you that your manuscript has been judged scientifically suitable for publication and will be formally accepted for publication once it meets all outstanding technical requirements.

Kind regards,

Daniel Oladimeji Oluwayelu, D.V.M., M.Sc., Ph.D.

Academic Editor

PLOS ONE
---

## [Editor Report · Acceptance letter]

13 Jun 2024

PONE-D-23-34199R2 

PLOS ONE

Dear Dr. Salu, 

I'm pleased to inform you that your manuscript has been deemed suitable for publication in PLOS ONE. Congratulations! Your manuscript is now being handed over to our production team.

Kind regards, 

on behalf of

Professor Daniel Oladimeji Oluwayelu 

Academic Editor

PLOS ONE